# NCodR: A multi-class support vector machine classification to distinguish non-coding RNAs in Viridiplantae

Chandran Nithin[1,2] ⓘ, Sunandan Mukherjee[1,3] ⓘ, Jolly Basak[4] and Ranjit Prasad Bahadur[1] ⓘ

[1]Computational Structural Biology Lab, Department of Biotechnology, Indian Institute of Technology, Kharagpur 721302, India; [2]Laboratory of Computational Biology, Faculty of Chemistry, Biological and Chemical Research Centre, University of Warsaw, 02-089 Warsaw, Poland; [3]Laboratory of Bioinformatics and Protein Engineering, International Institute of Molecular and Cell Biology, PL-02-109 Warsaw, Poland; [4]Department of Biotechnology, Visva-Bharati, Santiniketan, 731235, India

## Original Research Article

**Keywords:**
k-mer repeats; ncRNA prediction; non-coding RNA; RNA folding; SVM classifier.

C.N. and S.M. contributed equally and also considered as the joint first authors.

**Author for correspondence:**
R. P. Bahadur,
E-mail: r.bahadur@bt.iitkgp.ac.in

### Abstract

Non-coding RNAs (ncRNAs) are major players in the regulation of gene expression. This study analyses seven classes of ncRNAs in plants using sequence and secondary structure-based RNA folding measures. We observe distinct regions in the distribution of AU content along with overlapping regions for different ncRNA classes. Additionally, we find similar averages for minimum folding energy index across various ncRNAs classes except for pre-miRNAs and lncRNAs. Various RNA folding measures show similar trends among the different ncRNA classes except for pre-miRNAs and lncRNAs. We observe different k-mer repeat signatures of length three among various ncRNA classes. However, in pre-miRs and lncRNAs, a diffuse pattern of k-mers is observed. Using these attributes, we train eight different classifiers to discriminate various ncRNA classes in plants. Support vector machines employing radial basis function show the highest accuracy (average F1 of ~96%) in discriminating ncRNAs, and the classifier is implemented as a web server, NCodR.

## 1. Introduction

RNA molecules are involved in various cellular functions by catalysing biological reactions, controlling gene expression, or sensing and communicating responses to cellular signals (Morris & Mattick, 2014). On the one hand, the coding RNAs are translated into proteins by the cellular machinery of ribosomes. On the other hand, non-coding RNAs (ncRNAs) are transcribed but not translated and play significant roles in regulating gene expression (Cech & Steitz, 2014). Viridiplantae is a clade consisting of green algae and embryophytic land plant species. Identifying and classifying the ncRNAs in various plant species are important to study and understand their roles in regulating gene expression at various levels. In the current study, we analyse the various sequence and secondary structure-based folding measures of seven major classes of ncRNAs: long non-coding RNAs (lncRNAs), microRNAs (miRNAs), precursor-miRNAs (pre-miRs), ribosomal RNAs (rRNAs), small nucleolar RNAs (snoRNAs), transfer RNAs (tRNAs) and small nuclear RNAs (snRNAs).

The sequence length, AU content, minimum folding energy index (MFEI), normalised base-pairing distance (ND), normalised base-pairing propensity (Npb), normalised Shannon entropy (NQ) and k-mers of length three across the different classes of ncRNAs are analysed. These attributes are then used to train multiple classifiers to distinguish between the various classes of ncRNAs. The length of the sequence (L) and the AU content are two intrinsic properties of ncRNAs that determine its ability to form various secondary structures through canonical and non-canonical base pairing. The MFEI is the energy associated with RNA secondary structure formation normalised per GC content per 100 nucleotides (nt). The Npb quantifies the extent of base-pairing in the secondary structure of RNA. Npb values can range from 0.0, signifying no base pairs, to 0.5, signifying L/2 base pairs (Matera et al., 2007; Schultes et al., 1999). The normalised Shannon

entropy (NQ) is a measure of the conformational entropy of the secondary structure calculated from the MaCaskill base pair probability distribution (BPPD) (Huynen et al., 1997). The ND measures the base pair distance for all the pairs of structures (Huynen et al., 1997; Moulton et al., 2000). Previous reports have suggested that repeats of length one to six nucleotides play significant roles in functions of ncRNAs (Chen et al., 2010; Hazra et al., 2017; Jaiswal et al., 2020; Joy et al., 2013; 2018; Joy & Soniya, 2012; Mondal & Ganie, 2014; Sharma et al., 2021; Tabkhkar et al., 2020). The RNA folding measures and di-nucleotide compositions were previously employed to develop classifiers for ncRNAs (Barik & Das, 2018; Panwar et al. 2014). Moreover, in our previous studies, we used the k-mers of three nucleotides length to efficiently predict miRNAs (Nithin et al., 2015; 2017; Patwa et al., 2019).

In this study, we have used these attributes to develop eight classifiers. The support vector machine (SVM) employing radial basis function (RBF) shows the highest accuracy (96%) in discriminating ncRNAs and very high sensitivity and specificity of 0.96 and 0.99, respectively, for the average of 5-fold cross-validation experiments. The classifier developed in this study outperforms existing methods: RNACon (Panwar et al. 2014), nRC (Fiannaca et al., 2017) and ncRDeep (Chantsalnyam et al., 2020) for the sequences from Viridiplantae. The classifier will help advance our knowledge of ncRNAs by enhancing their genome-wide discovery and classification in various plant species. Moreover, a detailed understanding of ncRNAs will improve our knowledge of gene regulation at the transcriptional and post-transcriptional levels. These advances may further help develop better crop varieties with improved yield, productivity and better abiotic stress tolerance and disease resistance through genome editing technology (Basak & Nithin, 2015). The classifier is freely provided as a web server, NCodR (http://www.csb.iitkgp.ac.in/applications/NCodR/index.php), as well as a standalone tool (https://gitlab.com/sunandanmukherjee/ncodr.git).

## 2. Materials and methods

### 2.1. Dataset of ncRNAs

A dataset of ncRNAs in Viridiplantae was curated to quantify the various sequence and RNA folding measures, which is further used to classify ncRNAs. The dataset was curated from RNACentral (The Rnacentral Consortium, 2019), applying the filters on the ncRNA class and species names. The RNACentral consolidates data from several databases. While curating the dataset, the sequences with degenerate letters for the bases and ambiguous letters such as 'N' were removed. Sequences which repeated multiple times in the dataset were also removed to make them unique. We used a total of 526,552 sequences in further analysis (Table 1). The phylogenetic classification of the species was retrieved using NCBI taxonomy browser (Schoch et al., 2020). The dataset includes sequences from 46,324 species, which includes 40,219 vascular plants, 2,060 mosses, 1,885 green algae, 1,399 liverworts and 327 other green plants (Table 2). The 22,886 lncRNA sequences belong to 27 different species with eight monocots and 19 eudicots. The 9,328 miRNAs belong to 102 species while 118,912 pre-miRs belong to 883 species. The snoRNA sequences are from 114 species while snRNA sequences are from 110 species. The rRNA sequences are from 36,339 species while the tRNAs are from 15,170 species. In addition, we curated a dataset of 17,026 mRNAs which is used as 'others' category in this study. The mRNAs from 271 different species were downloaded from PlantGDB database (Dong et al.,

2004) and were clustered at 50% identity cut-off using CD-hit program (Li & Godzik, 2006).

### 2.2. RNA folding measures

The secondary structures of the RNAs were calculated using the RNAfold program (Hofacker et al., 1994). The MFEI value for a sequence of length $L$ was calculated using the adjusted MFE (AMFE), representing the MFE for 100 nt.

$$MFEI = \frac{AMFE}{(G+C)\%}.$$

$$AMFE = -\frac{MFE}{L} \times 100.$$

The genRNAstats program (Loong et al., 2006) was used to calculate NQ, ND and Npb for all the plant ncRNAs. Npb (Schultes et al., 1999) was calculated to quantify the total number of base pairs per length of the RNA sequence. The MaCaskill base-pair probability $p_{ij}$ between bases $i$ and $j$ was calculated using the equations below. Both the parameters, ND and NQ, were calculated using BPPD per base of the sequence (Huynen et al., 1997; Moulton et al., 2000).

$$NQ = -\frac{1}{L}\sum_{i<j} p_{ij}\log_2(p_{ij}).$$

$$ND = \frac{1}{L}\sum_{i<j} p_{ij}(1-p_{ij}).$$

$$p_{ij} = \sum_{S_\alpha \in S(s)} P(S_\alpha)\delta_{ij}.$$

$$P(S_\alpha) = \frac{e^{\frac{-E_\alpha}{RT}}}{\sum_{S_\alpha \in S(s)} e^{\frac{-E_\alpha}{RT}}}.$$

$$\delta_{ij} = \begin{cases} 1, & x_i \text{ pairs } x_j, \\ 0, & \text{otherwise.} \end{cases}$$

For each of the parameters, the probability distributions were computed as a Gaussian kernel density estimate using gnuplot (Racine, 2006).

### 2.3. k-mers of length three

Sequences were scanned for the presence of all the 64 k-mers of length three nucleotides. The sequence length in the dataset varies widely, and to remove this effect, the k-mer count was normalised per 100 nt ($R$) by the following equation:

$$R = \frac{\text{Number of k} - \text{mer signatures}}{L} \times 100.$$

We calculated the k-mer matrix of window size three for all the sequences.

### 2.4. Training and testing of classifiers

We train multiple classifiers using the dataset of ncRNAs to discriminate between the various classes to choose the most efficient classification algorithm. The classifiers were implemented in Python using the scikit-learn module (36). The following classifiers were trained to compare the performance measures: SVM using RBF kernel, Nearest Neighbours, SVM with linear kernel, Decision

**Table 1.** Dataset of Viridiplantae ncRNAs

| Parameter/RNA type | lncRNA | miRNA | pre-miR | rRNA | snoRNA | tRNA | snRNA |
|---|---|---|---|---|---|---|---|
| No. of unique sequences | 22,886 | 9,328 | 1,18,912 | 1,77,500 | 75,665 | 88,111 | 34,250 |
| Length (nt) | | | | | | | |
| Range | 200–24,018 | 15–29 | 55–11,032 | 60–9,601 | 19–1,854 | 26–677 | 18–912 |
| Mean ± sd. | 838 ± 955 | 21 ± 2 | 145 ± 103 | 733 ± 895 | 107 ± 25 | 75 ± 36 | 77 ± 67 |
| Median | 565 | 21 | 132 | 182 | 107 | 73 | 25 |
| Skewness | 5.80 | −0.36 | 20.52 | 1.61 | 21.22 | 8.91 | 0.85 |
| Kurtosis | 65.72 | 3.80 | 1,285.63 | 5.15 | 1,318.64 | 102.70 | 4.87 |
| AU (%) | | | | | | | |
| Range | 17.7–99.8 | 0.0–95.0 | 15.9–98.8 | 15.3–90.0 | 13.3–88.3 | 17.9–100 | 4.8–100 |
| Mean ± sd. | 59.2 ± 7.7 | 55.2 ± 12.1 | 62.3 ± 6.9 | 48.4 ± 5.5 | 59.3 ± 5.6 | 49.2 ± 7.1 | 54.4 ± 12.4 |
| Median | 60.64 | 55.0 | 63.4 | 47.9 | 59.8 | 48.6 | 54.9 |
| Skewness | −1.07 | −0.11 | −1.70 | 0.46 | −0.77 | 0.01 | −0.01 |
| Kurtosis | 4.55 | 3.63 | 9.37 | 4.22 | 4.70 | 0.02 | 0.03 |
| MFEI | | | | | | | |
| Range | 0.00–1.86 | 0.00–1.24 | 0.09–27.9 | 0.0–1.99 | 0.00–2.00 | 0.00–12.7 | 0.00–3.28 |
| Mean ± sd. | 0.59 ± 0.10 | 0.10 ± 0.13 | 1.39 ± 0.37 | 0.61 ± 0.10 | 0.48 ± 0.13 | 0.58 ± 0.47 | 0.38 ± 0.39 |
| Median | 0.60 | 0.03 | 1.42 | 0.61 | 0.46 | 0.60 | 0.38 |
| Skewness | 0.02 | 1.66 | 2.95 | −0.05 | 1.54 | 93.20 | 61.4 |
| Kurtosis | 0.03 | 6.41 | 228.15 | 6.28 | 13.32 | 9,773.20 | 7,886.88 |
| ND | | | | | | | |
| Range | 0.00–0.29 | 0.00–0.18 | 0.00–0.31 | 0.00–0.24 | 0.00–0.25 | 0.00–0.24 | 0.00–0.26 |
| Mean ± sd. | 0.12 ± 0.03 | 0.06 ± 0.04 | 0.03 ± 0.03 | 0.11 ± 0.04 | 0.10 ± 0.04 | 0.08 ± 0.04 | 0.07 ± 0.04 |
| Median | 0.12 | 0.05 | 0.02 | 0.11 | 0.10 | 0.08 | 0.06 |
| Skewness | 0.06 | 0.46 | 2.41 | −0.05 | 0.20 | 0.41 | 0.49 |
| Kurtosis | 3.01 | 2.52 | 11.04 | 2.76 | 2.31 | 2.54 | 2.81 |
| Npb | | | | | | | |
| Range | 0.00–0.46 | 0.00–0.41 | 0.11–0.49 | 0.07–0.48 | 0.00–0.47 | 0.00–0.47 | 0.00–0.44 |
| Mean ± sd. | 0.29 ± 0.03 | 0.11 ± 0.11 | 0.38 ± 0.04 | 0.31 ± 0.03 | 0.27 ± 0.04 | 0.29 ± 0.06 | 0.12 ± 0.12 |
| Median | 0.30 | 0.13 | 0.39 | 0.32 | 0.27 | 0.30 | 0.25 |
| Skewness | −1.24 | 0.32 | −0.85 | −1.35 | −0.29 | −1.60 | −0.84 |
| Kurtosis | 6.68 | 1.66 | 4.78 | 7.14 | 3.28 | 6.73 | 2.43 |
| NQ | | | | | | | |
| Range | 0.01–1.09 | 0.00–0.55 | 0.00–0.88 | 0.00–0.84 | 0.00–0.86 | 0.00–0.81 | 0.00–0.87 |
| Mean ± sd. | 0.41 ± 0.11 | 0.02 ± 0.10 | 0.08 ± 0.08 | 0.35 ± 0.13 | 0.32 ± 0.13 | 0.24 ± 0.12 | 0.21 ± 0.13 |
| Median | 0.40 | 0.16 | 0.05 | 0.35 | 0.31 | 0.23 | 0.19 |
| Skewness | 0.14 | 0.38 | 2.83 | −0.01 | 0.29 | 0.54 | 0.71 |
| Kurtosis | 3.08 | 2.52 | 13.61 | 2.76 | 2.46 | 2.88 | 3.36 |

Abbreviations: MFEI, minimum folding energy index; ND, normalised base-pairing distance; Npb, normalised base-pairing propensity; NQ, normalised Shannon entropy.

Tree, Random Forest, AdaBoost, Naïve Bayes and Quadratic Discriminant Analysis (Adankon & Cheriet, 2015; Breiman, 2001; Cover, 1965; Cover & Hart, 1967; Freund & Schapire, 1997; Knerr et al., 1990; Rivest, 1987). Additionally, we trained and tested a meta-classifier with a 'voting' approach using the average probability calculated by each of the aforementioned classifiers with equal weights for the prediction. The classifiers were trained using the following parameters: k-mer matrix, AU content, sequence length, MFEI, ND, Npb and NQ. The training dataset was made by randomly selecting 50% of the total sequences. We used the rest of the sequences for testing the classifier. We calculated the sensitivity or true positive rate, specificity or true negative rate, positive predictive value (PPV), negative predictive value (NPV), false-positive rate, false-negative rate, false discovery rate, accuracy and F score for the prediction from true positives, true negatives, false positives and false negatives using the following equations

**Table 2.** Taxonomy of dataset of Viridiplantae ncRNAs

| Taxon | lncRNA | miRNA | pre-miR | rRNA | snoRNA | tRNA | snRNA | All |
|---|---|---|---|---|---|---|---|---|
| Viridiplantae | 27 | 102 | 883 | 36,339 | 114 | 15,170 | 110 | 46,320 |
| | | | | | | | | | |
| |–Vascular | 27 | 93 | 850 | 31,596 | 98 | 13,096 | 93 | 40,219 |
| | | | | | | | | | | |
| | |–Seed | 27 | 92 | 849 | 31,140 | 97 | 12,452 | 92 | 38,226 |
| | | | | | | | | | | | |
| | | |–Flowering | 27 | 88 | 830 | 30,618 | 96 | 12,093 | 91 | 37,628 |
| | | | | | | | | | | | | |
| | | |–Monocots | 8 | 16 | 167 | 6,912 | 28 | 3,120 | 27 | 8,662 |
| | | | | | | | | | | | | |
| | | |–Eudicots | 19 | 68 | 656 | 22,435 | 65 | 8,426 | 61 | 27,416 |
| | | | | | | | | | | | | |
| | | |–Other flowering | 0 | 4 | 7 | 1,271 | 3 | 547 | 3 | 1,550 |
| | | | | | | | | | | | |
| | |–Other seed | 0 | 4 | 19 | 522 | 1 | 359 | 1 | 598 |
| | | | | | | | | | | |
| |–Ferns | 0 | 0 | 0 | 295 | 0 | 567 | 0 | 681 |
| | | | | | | | | | | |
| |–Club mosses | 0 | 1 | 1 | 161 | 1 | 77 | 1 | 203 |
| | | | | | | | | | |
| |–Mosses (Bryophyta) | 0 | 1 | 1 | 1,616 | 1 | 1,030 | 2 | 2,060 |
| | | | | | | | | | |
| |–Green algae (Chlorophyta) | 0 | 1 | 7 | 1,737 | 14 | 263 | 14 | 1,885 |
| | | | | | | | | | |
| |–Liverworts (Hepaticophyta) | 0 | 1 | 19 | 1,090 | 1 | 781 | 1 | 1,399 |
| | | | | | | | | | |
| |–Other green plants | 0 | 6 | 6 | 300 | 0 | 0 | 0 | 327 |

(Fawcett, 2006). We selected the best classifier for discriminating the ncRNAs based on the following performance measures:

$$\text{Sensitivity } (TPR) = \frac{TP}{TP+FN},$$

$$\text{Specificity } (TNR) = \frac{TN}{TN+FP},$$

$$PPV = \frac{TP}{TP+FP},$$

$$NPV = \frac{TN}{TN+FN},$$

$$FPR = \frac{FP}{FP+TN},$$

$$FNR = \frac{FN}{TP+FN},$$

$$FDR = \frac{FP}{TP+FP},$$

$$\text{Accuracy} = \frac{TP+TN}{TP+FP+FN+TN},$$

$$F = \frac{2 \times TP}{2 \times TP+FP+FN}.$$

The classifier implemented in the webserver was trained using ThunderSVM (Wen et al., 2018), which supports both GPU and CPU-based implementations, and is faster than Scikit learn-based SVM module. Moreover, the ThunderSVM algorithm is efficiently able to handle large sizes of the dataset as compared to the Scikit learn package. We have performed 5-fold cross-validation of the classifier with non-overlapping test sets. During the 5-fold cross-validation, the entire dataset is divided into five equal parts (20% each). During each iteration, a classifier is trained using four parts (80%) and the fifth (20%) is used for the validation. Each of these five experiments uses one fraction (20%) of the dataset for valida-tion in such a way that they do not overlap between the experi-ments. The average of the 5-fold cross-validation experiments has an accuracy of 96%.

## 3. Results and discussion

### 3.1. Length and AU content of the sequence

RNA molecules play significant roles in many biological processes by interacting with other RNAs, proteins or DNAs. In the recognition process, the length of RNA molecules varies from a few nucleotides in small ncRNAs to thousands of nucleotides in rRNA. The different classes of ncRNAs analysed in this study show

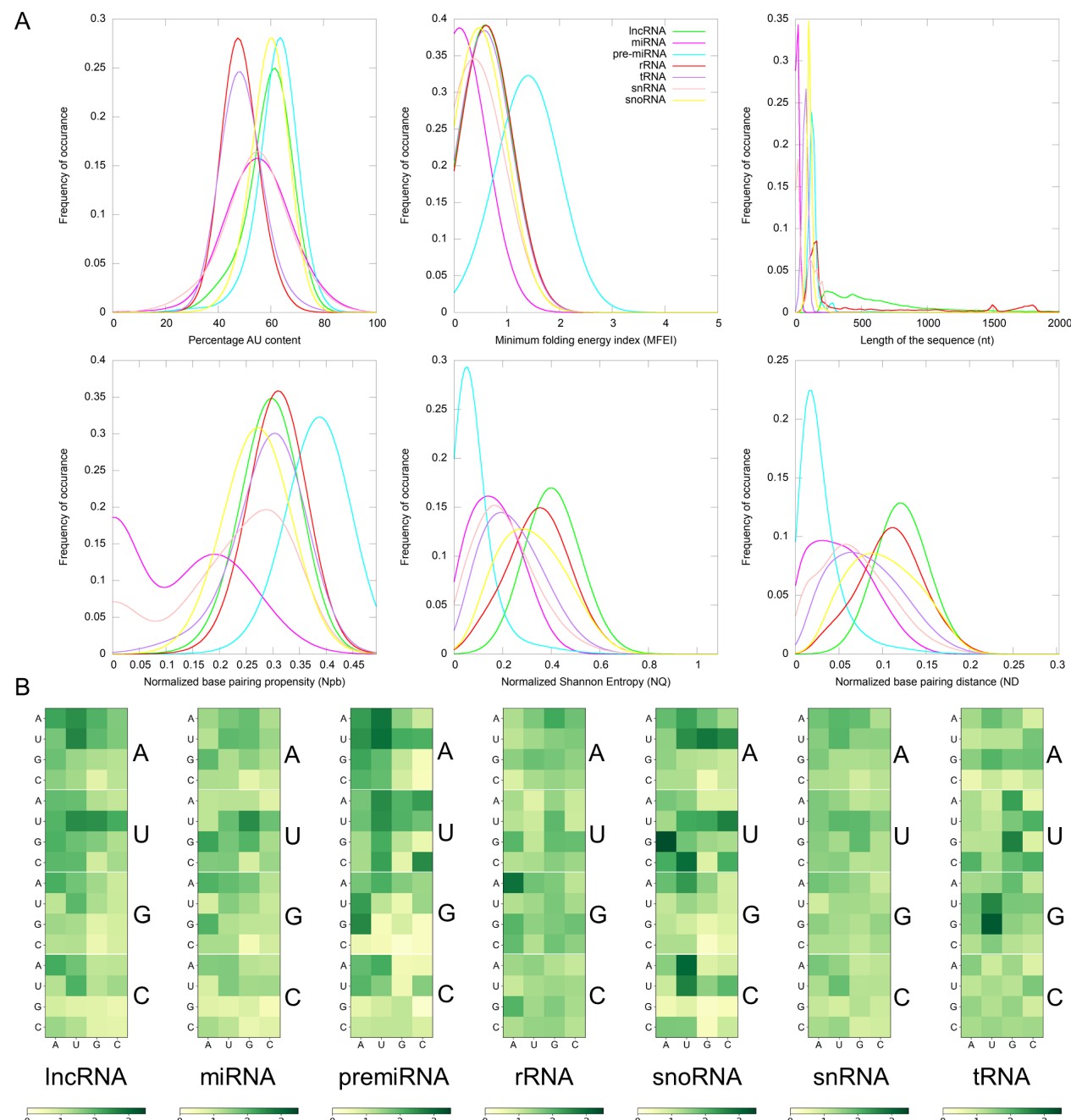

**Fig. 1.** (a) Distribution of Gaussian kernel density estimates of sequence length, percentage of AU content, MFEI, ND, Npb and NQ of different classes of ncRNAs in Viridiplantae. (b) Distribution of k-mers of length three in different classes of ncRNAs: k-mer signatures are represented in a nucleotide matrix of triplets representing four nucleotides A, U, C, G and 64 k-mers.

different length distribution, with miRNAs having the lowest mean length of 21 nt and lncRNAs having the highest mean of 838 nt. (Figure 1). The length of lncRNAs varies from 200 to 24,018 nt with a median of 565 nt. Among the various classes of ncRNAs analysed in this study, the highest deviation in length is observed for lncRNAs. The length of miRNAs varies from 15 to 29 nt with a median length of 21 nt. In pre-miRs, the length varies from 55 to 11,032 nt with a mean of 145 nt and a median of 132 nt. The length of rRNAs varies from 60 to 9,601 nt with a mean and median of 733 and 182 nt, respectively (Table 1). For the snRNAs, the length varies from 18 to 912 nt with a mean and median of 77 and 25 nt, respectively. The length of snoRNAs varies from 19 to 1,854 nt with

both mean and median values of 107 nt. The length of tRNAs varies from 26 to 677 nt with mean and median values of 75 and 73 nt, respectively. Among the various classes of ncRNAs, tRNAs show the lowest standard deviation in their length distribution.

Previous studies have shown that AU-rich elements (ARE) on mRNAs are recognised by ncRNAs, especially miRNAs, to induce mRNA decay (Jing et al., 2005). However, the exact mechanism of how AREs are recognised and processed by ncRNAs is still elusive. Significantly, the ncRNAs have distinct AU content than that of coding RNAs, and we analyse their distribution among the various ncRNA classes. The mean AU content for different ncRNA types and their range and standard deviation are shown in Table 1.

Among the various classes of ncRNAs, we observe that the AU content in both pre-miRs and lncRNAs has the widest variation, ranging from 16 to 99%. The highest mean and median values are observed for pre-miRs with 62% and 63% values, respectively, while lncRNAs have mean and median values at 59% and 60%, respectively. The snRNAs have a mean AU content of 54% and a median value of 55%, while snoRNAs have mean and median values at 60%. The lowest mean and median values of AU content are observed in both tRNAs (mean 49%; median 49%) and rRNAs (mean 49%; median 48%). Figure 1 shows the distribution of AU content among the various classes of ncRNAs. The distributions of AU content are distinct among the different classes of ncRNAs, yet there exist considerable overlapping regions across all the classes.

## 3.2. RNA folding measures

The various ncRNA classes have distinct secondary structures, quantified by calculating the various RNA folding measures, including MFEI, ND, Npb and NQ. The MFEI values represent the normalised free energy associated with the formation of the secondary structure. The MFEI values were calculated for the different classes of ncRNAs, and it is observed to have similar mean values across various classes except for pre-miRs and lncRNAs. Despite the similar mean values, the range of MFEI varies across the different classes. The MFEI varies from 0.00 to 1.98 in rRNAs, 0.00 to 2.00 in snoRNAs, 0.00 to 12.70 in tRNAs and 0.00 to 3.28 in snRNAs (Table 1). The pre-miRs show the highest standard deviation with MFEI ranges between 0.00 and 27.90, while the lncRNAs have the lowest standard deviation with ranges between 0.00 and 1.86. The probability distribution of MFEI among various ncRNAs classes shows a distinct non-overlapping region for pre-miRs and overlapping areas with other ncRNAs (Figure 1).

A clear distinction is observed in the distributions of ND values of pre-miRs and lncRNAs compared to the other classes of ncRNAs (Figure 1). The mean ND value is lowest (0.02) for pre-miRs, which ranges from 0.00 to 0.31 (Table 1). The highest mean (0.12) is observed for lncRNAs, and it ranges from 0.00 to 0.29. The rRNAs have a mean ND value of 0.11, with a range from 0.00 to 0.24. The snoRNAs, tRNAs and snRNAs have mean ND values of 0.10, 0.08 and 0.07, respectively, with similar ranges of 0.0–0.25, 0.00–0.24 and 0.00–0.26, respectively. Figure 1 shows the pre-miRs have a skewed distribution towards the lower ND values, with 99% of the population lying below 0.15. However, the distribution is symmetric for both lncRNAs and rRNAs (Figure 1).

The Npb values for no base pairing between the nucleotides to complete base pairing can range from 0.0 to 0.5. Here, the base pairing signifies the internal base pairs formed by the RNA as part of the secondary structure of the molecule. We observe that the mean values of Npb across different ncRNA types are similar, except for a slightly higher value for pre-miRs (Table 1). However, the range widely varies across the different classes. In the case of lncRNAs, base pairing is observed for 0 to 92% of nucleotides with a 60% mean. The highest base pairing is observed in pre-miRs, ranging from 21 to 99%, with a mean of 77%. For rRNAs, tRNAs snRNAs and snoRNAs, the base pairing ranges from 14 to 97%, 0 to 94%, 0 to 87% and 14 to 94% with mean values of 60, 58, 43 and 54%, respectively. Except for pre-miRs, the probability distribution of Npb values shows that the different ncRNA classes have distributions centred around the similar mean values (Figure 1). Besides, it shows that the distribution of pre-miRs shifts towards higher base-pairing regions compared to other classes of ncRNAs.

RNA molecules, being more flexible than proteins (Bahadur et al., 2009; Mukherjee & Bahadur, 2018; Nithin et al., 2017), have higher conformational freedom levels than proteins. This conformational freedom of the RNA is measured in terms of Shannon entropy. The secondary structure is calculated using the RNAfold program which uses the equilibrium partition function and the BPPD to determine the minimum folding energy structure (McCaskill, 1990). An ensemble of permissible RNA secondary structures, represented using MaCaskill BPPD, provides the probability values for base pairing at each position in a sequence (Huynen et al., 1997). Shannon entropy (Shannon, 1948) was calculated using the base-pairing probability and normalised per length of the sequence to obtain the NQ values. A lower value signifies that the distribution is dominated by a single or a few base-pairing probabilities, while a higher signifies the possibility of alternative conformations (Huynen et al., 1997; Schultes et al., 1999). The highest average value of NQ is observed for lncRNAs that range from 0.01 to 1.09, with a mean of 0.41. The lowest NQ observed is for miRNAs followed by pre-miRs, and it ranges from 0.00 to 0.55 and 0.00 to 0.88, with a mean of 0.02 and 0.08, respectively (Table 1). The rRNA class shows NQ values ranging from 0.00 to 0.84, with a mean of 0.35. The snoRNAs, tRNAs and snRNAs have NQ values ranging from 0.00 to 0.86, 0.00 to 0.81 and 0.00 to 0.87, with mean values of 0.32, 0.24 and 0.21, respectively. NQ values' probability distributions show different skewness (Table 1) across different ncRNA classes with exceptions for rRNAs with the symmetrical distribution centred on the mean (Figure 1). We observe the highest degrees of both skewness and kurtosis in pre-miRs.

## 3.3. k-mers of length three

In this study, we have checked the presence of k-mers of three nucleotide length in different classes of ncRNAs. In our previous studies, we have shown that k-mers of three nucleotides are conserved across the same miRNA family (Nithin et al., 2015; 2017). On an average, the k-mer signatures UUA, AAG, CUU, UGA, GAA, UUC, CAA, AAA, UUG, AAU, UUU and AUU are repeated at least two times per 100 nucleotides in lncRNAs, while AUG is repeated 1.95 times. AUG acts as the start codon. Additionally, signatures, UUG, AAG and AUU are reported to act as alternate start codons in protein synthesis (Kearse & Wilusz, 2017). These sequence motifs can be attributed to the fact that the lncRNAs usually co-locate on the genome and the protein-coding genes in sense or antisense directions (Ziegler & Kretz, 2017). The presence of start codons in lncRNAs is expected as ncRNAs often contain several potentially translated ORFs (Ingolia et al., 2011). Also, the poor conservation of lncRNAs across species means that the small ORFs in lncRNAs, if translated, will produce species- or lineage-specific peptides (Ruiz-Orera et al., 2014). Moreover, the de novo protein-coding gene evolution is theorised to be a continuum from non-functional genomic sequences to fully fledged protein-coding genes (Carvunis et al., 2012). The lowest average R-value in lncRNA is observed for the k-mer CGC. Moreover, we did not observe any significant conserved k-mers across all the lncRNAs as well as in pre-miRs. Although the k-mers are not conserved across various miRNA families (Kozomara et al., 2019), they are conserved within each of the different miRNA families (Nithin et al., 2015). These conserved k-mer signatures were utilised previously to remove the false positives in the prediction of miRNAs (Nithin et al., 2015; 2017; Patwa et al., 2019). The general trend observed both in lncRNAs and pre-miRs is the presence of a diffuse pattern across

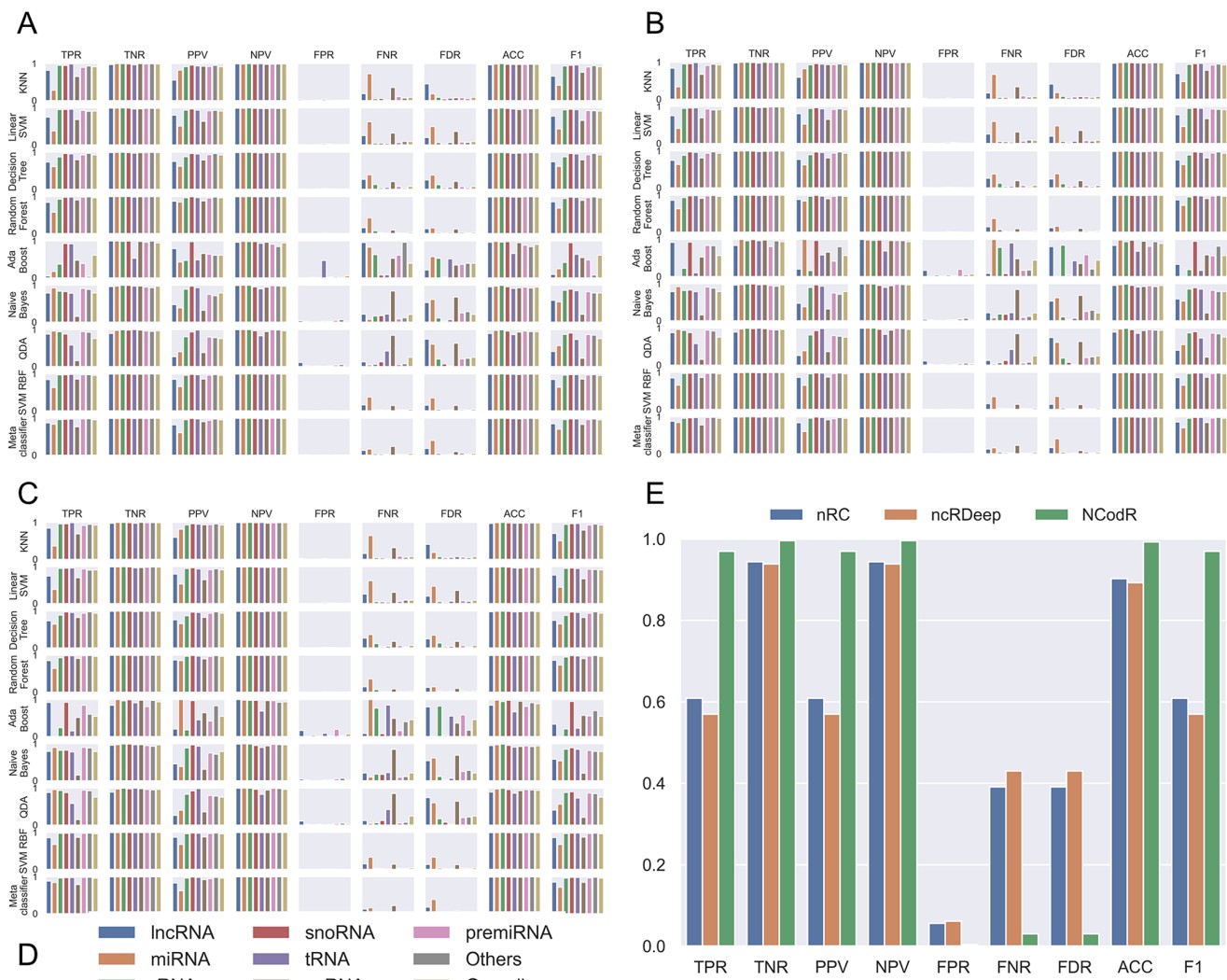

**Fig. 2.** Performance measures for different classifiers. The classifiers were trained using scikit-learn with different proportions of training/testing datasets: (a) 50/50, (b) 80/20 and (c) 95/5. The legends (d) are common for the first three panes. (e) For the test dataset with Viridiplantae non-coding RNAs, NCodR shows significantly higher performance than nRC and ncRDeep methods.

various k-mer signatures. A total of five signatures (UGA, GGA, CGA, AAG, GAA) are observed in rRNAs to have a mean $R$ of 2.0 or more with the highest value of 2.90 observed for GAA. The highest frequency (3.36) of any k-mer observed in ncRNAs is for UGA in snoRNAs. The signature CAU also shows a high mean $R$ of 3.07 in snoRNAs, where 12 other k-mers also have $R$ values more than 2.0. GGU has the highest frequency in tRNAs ($<R> = 3.23$) with seven other signatures (AGU, UCA, UUC, UCC, UAG, GUU, UGG) having R values more than 2.0. We find only one k-mer with mean $R$ more than 2.0 in snRNAs, AUU ($<R> = 2.02$). The distribution of all 64 k-mers among various classes is shown in Figure 2. Each class of ncRNA shows a unique pattern of k-mers, making it an excellent parameter for their discrimination. Moreover, previous studies suggest that k-mer-based classification has proved to be a powerful approach in detecting relationships between sequence and function in lncRNAs (Kirk et al., 2018; 2021).

### 3.4. Classifier to discriminate ncRNA types

Using the various sequence-based parameters and RNA folding measures developed in this study, we trained eight classifiers and

compared their sensitivity, specificity, PPV, NPV and F score (Figure 2). The Nearest Neighbours algorithm was trained using a k value of ten and KD tree algorithm. The Linear SVM was trained with the penalty parameter C set to 1. The Decision Tree classifier was run without constraining the depth of the trees. Both Random Forest and AdaBoost classifiers were trained with the number of estimators set to 1,000. The multi-class SVM is implemented with RBF kernel (SVM-RBF) and is based on the 'one-against-one' approach. The kernel parameter $\gamma$ was set to $1/n$, where $n$ is the number of parameters. The penalty parameter C was empirically optimised by minimising the error rate on the training data by performing a grid search before the final training. A meta-classifier was trained as an ensemble of all the eight classifiers with identical parameters used for training of each of the individual classifiers. For all other classifiers, default parameters were used. Moreover, we trained and tested the classifiers with three different proportions of training/testing datasets: 50/50, 80/20 and 95/5.

Among the various classifiers, the overall performance of SVM-RBF, meta-classifier and Random Forest is surpassing. F-score, which represents the harmonic mean between precision and recall, is around 0.95 for all the three classifiers (Figure 2). Moreover, all

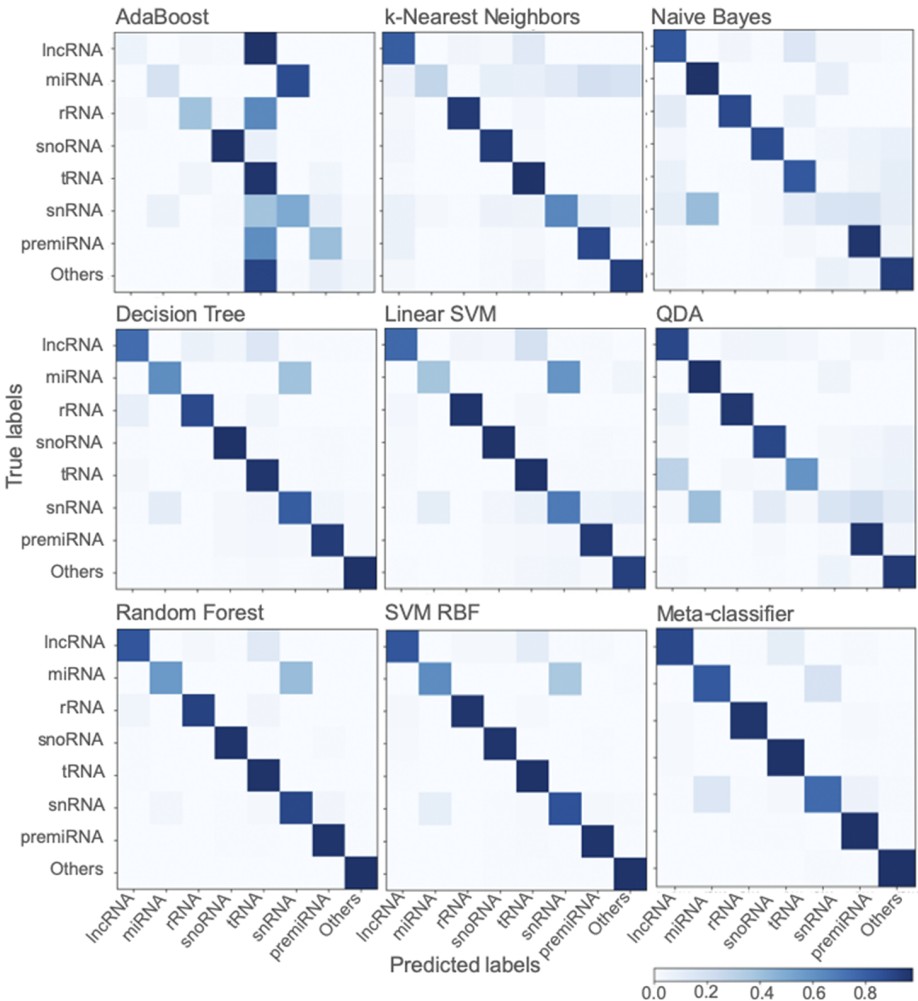

**Fig. 3.** Confusion matrix generated using different classifiers. The confusion matrix shows the performance of classifiers trained using scikit-learn using a dataset split of 50/50 for training and testing.

the three classifiers with all three sets of training/testing datasets have an accuracy of ~99% with overall sensitivity and specificity above 0.95 and 0.99, respectively (Figure 2). The confusion matrix for the 50/50 training/testing dataset is shown in Figure 3. For some of the ncRNA classes, meta-classifier shows better prediction accuracy than SVM-RBF or Random Forest, but it is several times slower than the other two classifiers. We performed five-fold cross-validation using 95% of the training data. It randomly shuffles and splits the dataset into five groups and randomly selects four groups for training and remaining one for testing with a ratio of 80/20, respectively.

The SVM-RBF classifier was optimised by testing different values for hyperparameters C and gamma. Different values of C from 1 to 50 were tested while gamma values ranging from 0.001 to 50 were tested. For each tested combination of C and gamma, we performed the training and testing of the classifier with an 80/20 split of dataset and calculated the performance measures. The optimum values for C and gamma were selected as 8 and 0.003, respectively (Supplementary Figures S1 and S2, Supplementary Material).

### 3.5. lncRNAs in diverse taxa

lncRNAs are a class of large, diverse, transcribed, non-protein coding RNA molecules with more than or equal to 200 nucleotides.

Since the first discovery of the regulatory role of lncRNA in an important biological process in the Drosophila bithorax complex (Lipshitz et al., 1987), they were identified in several plants and animals. The lncRNAs have certain specific properties including lower abundance, restriction to particular tissues or cells and less frequent conservation between species (Derrien et al., 2012). These specific properties make the prediction and identification of lncRNAs difficult. As a result, the number of lncRNAs available in the RNACentral database is limited to a handful of species. Even though the overall dataset of ncRNAs used to train the classifier is diverse in terms of the taxa included, the data for lncRNAs come only from two major groups: monocots and eudicots (Table 2). The limited availability of data from diverse taxonomic groups for the lncRNA class may cause the classifier to have limited predictive ability for lncRNAs. To test this, we curate an additional dataset of lncRNAs from four different species that were not included in the original training dataset. The four taxonomically defined, diverse species we have used to test the classifier are *Micromonas pusilla* (green algae/Chlorophyta), *Physcomitrella patens* (mosses/Bryophyta), *Galdieria sulphuraria* (red algae/Rhodophyta) and *Selaginella moellendorffii* (club-mosses). lncRNA sequences of *M. pusilla* and *S. moellendorffii* were downloaded from the GREENC database (Paytuví Gallart et al., 2016) while that of *G. sulphuraria* and *P. patens* were from

CANTATAdb (Szcześniak et al., 2016). Of the 651 lncRNAs of *M. pusilla* lncRNAs, 63% (410) were correctly classified. For the red algae, *G. sulphuraria,* 97.6% (1871) of the 1,917 lncRNAs were classified correctly. In the case of the mosses, *P. patens* 73.76% (1,498) were predicted correctly as lncRNAs. In the case of *S. moellendorffii,* 67.36% (1,483) of the 2,267 sequences were predicted as lncRNAs. In spite of the lack of data from these taxa in the training dataset, the classifier is able to predict the majority lncRNAs correctly. This demonstrates the robustness of the features used in training the classifier. With the availability of more lncRNA sequences in the future, the classifier could be retrained to improve the predictive power.

In order to further test the utility of the classifier on a wider scale, we downloaded the Plant Long non-coding RNA Database (PLncDB) (Jin et al., 2020) which consist of 1,008,006 lncRNA sequences from 80 different species. The classifier was able to predict 82.15% of the sequences correctly as lncRNAs. This demonstrates the utility of NCodR in the large-scale classification of plant ncRNAs with high accuracy.

The classifier is developed to efficiently discriminate the different classes of ncRNA, and it may not be very effective in differentiating coding RNAs from ncRNAs. The classifier performs best when the dataset is pre-screened to separate the coding and ncRNAs. For discriminating coding RNAs from ncRNAs, specialised tools such as Coding Potential Calculator (Kong et al., 2007) and Coding Potential Assessment Tool (Wang et al., 2013) are available. Moreover, the combination of these tools can also effectively discriminate coding RNAs from ncRNAs (Nithin et al., 2017).

### 3.6. Comparison with other classifiers

The SVM-RBF developed in this study was compared with nRC, ncRDeep and RNACon. In order to test and compare the classifiers, we use a dataset of 1,084 sequences. The dataset was prepared by randomly selecting 1% of sequences from each class of ncRNAs available in the testing dataset. The RNACon program generated output with sequences classified into coding and ncRNAs. RNAcon had predicted 90% of the ncRNAs correctly as non-coding; however, it failed to classify further into the different classes of RNAs. In the subsequent analysis RNAcon was excluded. The various performance measures were computed for nRC and ncRDeep and compared with NCodR. NCodR outperforms both the classifiers (Figure 2e) for predicting plant ncRNAs. The performance measures reported of NRC and ncRDeep are low for plant ncRNAs. This could be attributed to the fact that the original training and testing dataset did not include much of the plant ncRNAs. This observation is in line with the previous studies with miRNA prediction tools. The tools which were not developed specifically for plant miRNAs performed poorly and predicted very low number of miRNAs in *Phaseolus vulgaris* and *Cajanus cajan* which prompted us to adapt and develop the pipeline to be specific for plant miRNAs (Nithin et al., 2015; 2017).

### 3.7. Implementation and availability of the classifier

The classifier is implemented as a web server, NCodR, which is freely available at the URL: http://www.csb.iitkgp.ernet.in/applications/NCodR/index.php. Users need to submit the sequence in FASTA format, and the output generated by the NCodR shows the ncRNA class. For large-scale prediction, NCodR can be set up locally to use it as a 'standalone' tool. The tool is implemented using a combination of scripts developed in C++, Perl and Python. Details about installation and manual for usages are publicly available from the git repository (https://gitlab.com/sunandanmukherjee/ncodr.git). Additionally, NCodR is also available as a precompiled Docker image (https://hub.docker.com/repository/docker/nithinaneesh/ncodr). The dataset used in the study is available in Mendeley Data at the URL: http://doi.org/10.17632/87k9rssdm4.2.

## 4. Conclusions

In this study, based on the sequence and structural measures of known ncRNAs, we have developed eight classifiers. The multi-class SVM-RBF model was the best in discriminating the ncRNAs and is implemented in a web server, NCodR, to classify ncRNAs in Viridiplantae. The SVM-RBF has an F-score of 0.96, along with a specificity of 0.96 and a sensitivity of 0.99. This classifier can be further used for genome-wide identification and classification of ncRNAs in various plant species. Identification and classification of ncRNAs will improve our understating of gene regulation in plants at the transcriptional and post-transcriptional levels. An improved understanding of ncRNAs will help us develop better varieties of crops through genome-editing technology with improved yield, productivity, better stress tolerance and disease resistance. Furthermore, the classifier has significant predictive ability of classifying sequences in diverse taxonomic groups which will help us in the further advancing of knowledge of ncRNAs in general.

### Acknowledgements

The authors acknowledge Amal Thomas and Srinivasan Sivanandan for their valuable insights in the implementation of the program. C.N. thanks the Poznan Supercomputing and Networking Center, the ACK Cyfronet AGH and the Polish Grid Infrastructure PL-Grid for the computing grant (Grant ID: plgmodeling).

**Financial support.** C.N. thanks IIT Kharagpur for the fellowship and both EMBL Advanced Training Centre Corporate Partnership Programme and IIT Kharagpur for the travel grants to present this work at international conferences. C.N. acknowledges the grant from the National Science Centre, Poland (Grant ID: OPUS 2019/33/B/NZ2/02100). S.M. thanks UGC and IIT Kharagpur for the fellowship. J.B. thanks CSIR, India, and R.P.B. thanks DST, India.

**Conflict of interest.** The authors declare none.

**Authorship contributions.** C.N. and S.M. designed the study, performed the data analysis and the development of models, standalone and webserver. Both C.N. and S.M. contributed equally to this manuscript. J.B. and R.P.B. supervised the work. All authors participated in the writing of the manuscript.

**Data availability statement.** The dataset used in the study is available in Mendeley Data at the URL: https://doi.org/10.17632/87k9rssdm4.2. The source code, instructions for installation and the user manual are publically available from the git repository https://gitlab.com/sunandanmukherjee/ncodr.git. The webserver is publicly available at the URL: http://www.csb.iitkgp.ernet.in/applications/NCodR/index.php.

**Supplementary Materials.** To view supplementary material for this article, please visit http://doi.org/10.1017/qpb.2022.18.

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
