## [Reviewer Report]

June 25th, 2021

To,

The Editor,

Quantitative Plant Biology

Dear Prof. Hamant,

Thank you for your email dated 01 March 2021 inviting us to submit our manuscript entitled “NCodR: A multi-class SVM classification to distinguish between non-coding RNAs in Viridiplantae.” Please find enclosed our manuscript for publication as a research article in Quantitative Plant Biology.

Non-coding RNAs (ncRNA) are major players in the regulation of gene expression. In this study, we analyse seven classes of non-coding RNAs to identify the sequence and RNA folding based measures and use them to discriminate the various classes of ncRNAs in plants. We train eight different classifiers using these attributes. Among the various classifiers, the support vector machine-based classifier using the radial basis function has the best performance in discriminating ncRNAs. This study will provide a reliable platform for the genome-wide prediction and classification of ncRNAs in plants and enrich our understanding of plant ncRNAs, which may be further used for crop improvements using genome-editing methods. Our method combines the use of both RNA folding measures and simple sequence repeats to classify non-coding RNAs. The SVM-RBF trained in this study has an average F1 (a mix of precision and recall) of ~91%, with a specificity of 0.96 and sensitivity of 0.99. Large-scale classification of ncRNAs in plants is still missing in the literature, and this study bridges the gap. The algorithm is implemented both as a web server available at http://www.csb.iitkgp.ernet.in/applications/NCodR/index.php and as a standalone program available at https://gitlab.com/sunandanmukherjee/ncodr.git.

Large-scale classification of ncRNAs in plants is still missing in the literature, and we completely agree with you that Quantitative Plant Biology will be the best platform to publish this work. In this line, we believe that our method for discriminating ncRNAs will be of particular interest to the readers of Quantitative Plant Biology, especially for those trying to analyse ncRNA in whole genomes.

With regards,

Ranjit P. Bahadur,

Professor, Computational Structural Biology Lab

Department of Biotechnology

Indian Institute of Technology Kharagpur

Kharagpur-721302, India

---

## [Reviewer Report]

*Comments to Author*: In their manuscript, Nithin and al. used an algorithmic approach to help studying plant non-coding RNAs (ncRNAs) and developed an online tool (NCodR), freely usable to classify them, based on machine learning.

To achieve that, they first built a dataset of ncRNAs from a wide variety of plant species. They defined various attributes derived from the sequences of each known type of ncRNA (length, AU content, ability to form secondary structures, composition in k-mers of length 3…). Based on these attributes, they trained several commonly used classifiers: support-vector machines, decision trees, random forests, etc., and they evaluated the sensibility and specificity of each classifier on a validation set.

The code of the project is available from a public repository.

The overall process is well explained, notably the various attributes used as input data for the classifiers. Training and validation of the classifiers follow the best practices of the field. The choice of SVMs appears to be a good idea as, in addition to the good performances that the authors show, they allow the interpretation of the obtained hyperplanes, while more “fashionable” classifiers, like neural networks, act as black boxes. But the study would need some additional work to be really helpful to the field of plant biology.

To try the online interface of NCodR, I used a small set of RNA-seq data. The prediction was that 33 sequences out of 38 were lncRNAs, and 5 were rRNAs, while most of the input sequences were mRNAs (or at least the majority of them). I checked the predicted rRNA sequences by comparing them to the SILVA database (https://www.arb-silva.de/), and was not able to confirm any of them. I guess the problem is that messenger RNAs were not included in the training set. I can understand that the focus was put on ncRNAs, but the first real-life use I can see for NCodR is in reanalyzing total RNA libraries, to isolate ncRNAs from mRNAs and classify them. The second use would be to annotate genomes, but again, the software should be able to support having input sequences that are not genuine ncRNAs.

The authors should really give examples of real-life applications of NCodR.

Some attributes may not be relevant, depending on the data source. For example, the use of ‘U’ as a nucleotide is not standardized, and ‘T’ can be used instead. People use ‘U’ for sequences that have already been proved to be functional RNAs, but never in raw sequences or messenger RNAs.

The length of the sequences in the databases does not necessarily reflect the length of sequences that are experimentally obtained. Often raw sequences are in the range 100-150 nt due to sequencing limitations, even when they are derived form a lncRNA (expected to be longer than 200nt, Ziegler and Kretz, [60]). The length of the sequence is usually correct only after it has been curated, at a point when classification tools are not useful any longer.

So probably, depending on the applications the authors will suggest for NCodR, some attributes should be changed or removed from the dataset.

Notes:

L52: Viridiplantae are the *green* plant species, not all the plant species.

L251: Clarify the relationship between conformational freedom and Shannon entropy of the sequence. I’m not clear if “base-pairing” means di-nucleotides in the sequence or is related to self-hybridization of the molecule.

L271: “AUG acts as the start codon”: it is not expected that start codons are a frequent feature of non-coding RNAs. Please explain more.

L346: Red algae are not part of Viridiplantae (and are quite distant, in terms of evolution), so having NCodR correctly predict ncRNA categories for them show that the tool does not depend heavily on its training set and that ncRNA characteristics can be generalized. That’s a nice result on which the authors could put more stress.

---

## [Reviewer Report]

*Comments to Author*: Dear Prof. Bahadur,

Many thanks for submitting your manuscript QPB-21-0034 - "NCodR: A multi-class SVM classification to distinguish between non-coding RNAs in Viridiplantae" to Quantitative Plant Biology. My apologies for the long review process -- this has been in large part due to the difficulty of acquiring reviewers during the pandemic.

We have now heard back from two reviewers, who are broadly positive about your manuscript and online tool. However, they raised several concerns, which must be addressed before this manuscript can be published. One reviewer has additionally stress-tested the NCodR online interface and their experience has raised some points for improvement.

The most important classes of points are:

(a) absence of comparison to existing approaches from the literature (see attachment for some examples);

(b) relationship with what I will call the "family" of RNAs -- e.g. why not consider mRNAs, when distinguishing mRNAs and ncRNAs is likely an important usage; why are there so few miRNAs in the dataset; etc.

And each reviewer also raises specific questions, including the aforementioned stress test, the use of length for raw sequences, and whether the approach has original plant-specific elements. I believe addressing these questions will substantially strengthen the manuscript, and look forward to reading a revised version.

One review follows this text; the other should be automatically included by the review system.

Yours sincerely,

Iain Johnston

-----

The authors are interested in this paper by the classification of non-coding RNAs, proposing

classifiers based on SVM machine learning technique, and applied to particular species, the

Viridiplantae.

The classification of non-coding RNAs is an important task, which needs innovative tools.

Several methods and tools have been proposed in the literature. We can cite for instance:

nRC:

A. Fiannaca, M. La Rosa, L. La Paglia, R. Rizzo, A. Urso

Nrc: non-coding RNA classifier based on structural features

BioData Mining, 10 (2017), 10.1186/s13040-017-0148-2

RNAcon:

B. Panwar, A. Arora, G. Raghava

Prediction and classification of ncRNAs using structural information

BMC Genomics, 15 (2014), p. 127, 10.1186/1471-2164-15-127

ncRDeep:

Tuvshinbayar Chantsalnyam, Dae Yeong Lim, Hilal Tayara, Kil To Chong,

ncRDeep: Non-coding RNA classification with convolutional neural network,

Computational Biology and Chemistry, 88, (2020), 10.1016/j.compbiolchem.2020.107364

My main concern with this work is therefore the fact that none of these articles have been

cited, and no comparison with these already published methods has been made.

This is absolutely necessary before publishing any new method. The authors should justify

why their method is better, more innovative or more interesting than the published

methods, or more suitable for the considered species, here Viridiplantae, than the already

exiting methods.

My second concern is about the originality of the proposed method and why it is suitable for

Viridiplantae. The used features, features of sequence (including the well-known and very

used k-mer motifs) and of secondary structure, are generic and classical features for non-

coding RNAs. In any case, the authors did not present any feature as a specific feature of

these species.

It seems to me that the suitability of their method to Viridiplantae is only in the training data

used.

In general, there is an important lack of justifications on the method and the results.

Another point is about long non-coding RNAs (lncRNAs). It is limiting to consider them as a

one class, at the same level than the class of pre-miRNAs, the class or ribosomal RNAs,

transfert RNAs, etc. The lncRNAs are non-coding RNA with a size greater than 200

nucleotides, in opposite to the small non-coding RNAs like snoRNAs, miRNAs, siRNAs,

piRNAs, etc. And contrary to the small ncRNAs, the long ncRNAs as not well known yet, and

therefore not yet well defined and classified.

I have also a remark about the class of miRNAs. The authors said that the results were not

good, because the lack of data:“The poor performance is observed in the miRNA class, which can be attributed to fewer unique sequences in the dataset. “

This is very strange since the miRNAs represent, among the ncRNA classes, one of the most

studied class one, and for which there exist a lot of data…. ?

The authors set lots of performance scores in the text but use them only only in

Supplementary data. They should give figures summarizing these performances in the main

text (not only in Supplementary data) as well as a figure with comparisons with other tools

For the Figure 3, the authors did not detail which datasets they used to get these scores?

---

## [Reviewer Report]

April 23rd, 2022

To,

Dr. Olivier Hamant

Editor-in-Chief,

Quantitative Plant Biology

Dear Prof. Hamant,

Thank you very much for your email dated October 4, 2021 attaching the referees’ comments on our manuscript entitled “NCodR: A multi-class SVM classification to distinguish between non-coding RNAs in Viridiplantae” for revision. We apologize for the delay in addressing the reviewers' suggestions and thank you for extending the deadline multiple times. We also want to thank the reviewers for appreciating our work and for raising careful concerns towards the betterment of the study. I believe that incorporating these suggestions definitely improves the quality of the manuscript and hope it will be accepted by Quantitative Plant Biology. We have provided a point by point response to the referres comments and the relevant modifications/additions are coluored in red in the manuscript.

With regards,

Ranjit P. Bahadur,

Professor, Computational Structural Biology Lab

Department of Biotechnology

Indian Institute of Technology Kharagpur

Kharagpur-721302, India

---

## [Reviewer Report]

*Comments to Author*: In their manuscript, Nithin and al. used an algorithmic approach to help studying plant non-coding RNAs (ncRNAs) and developed a tool, NCodR, to classify them, based on machine learning. NCodR is freely available online, and can also be deployed as a Docker container.

To achieve that, they first built a dataset of ncRNAs and mRNAs from a wide variety of plant species. They defined various attributes derived from the sequences of each known type of ncRNA (length, AU content, ability to form secondary structures, composition in k-mers of length 3…). Based on these attributes, they trained several commonly used classifiers: support-vector machines, decision trees, random forests, etc., and they evaluated the sensibility and specificity of each classifier on a validation set.

The code of the project is available from a public repository on GitLab.

The paper had already been submitted to Quantitative Plant Biology journal in 2021. The new version has been improved, with better explanations of some attributes used as input data for the classifiers, a better assessment of the training/validation procedure and a comparison to similar tools (nRC, ncRDeep and RNACon).

The packaging as a Docker image is a great addition, as it allows to easily install the tool locally. Instructions on the Gitlab repository are very clear and useful. Note that one command should be changed on page https://gitlab.com/sunandanmukherjee/ncodr#212-pulling-the-image-from-dockerhub:

docker tag ncodr:latest nithinaneesh/ncodr:latest

should be

docker tag nithinaneesh/ncodr:latest ncodr:latest

The main application I can imagine for NCodR is to quickly sort RNA-seq reads into different categories, and extract classified ncRNA sequences. For this purpose, the authors have included 17 026 mRNA sequences in the training dataset, tagged as “others”. To test this application, I downloaded a random RNA-seq dataset from NCBI (SRR1588449), cleaned it, merged forward and reverse reads when possible, using PEAR (doi: 10.1093/bioinformatics/btt593) and ran NCodR on a subset of this dataset. From 100 049 reads, NCodR identified:

- 37 490 lncRNA,

- 29 871 rRNA,

- 23 025 premiRNA,

- 7 817 snoRNA,

- 1 511 snRNA,

- 319 tRNA,

- 16 mRNA.

That’s quite difficult to believe that an RNA-seq dataset that has been used for a gene expression paper contains only 0.016% of mRNA reads. On a small subset of these RNA-seq reads, I’ve tried to exclude the sequence length from the attributes used by the classifier, but it didn’t really improve the outcome. RNACon (Panwar et al, [46]) seems to be able to identify mRNAs (I haven’t tried the software, though).

I suggest to the author to try on another public dataset, to check the outcome I got with their tool.

Alternatively, they should suggest other real-life applications for NCodR, in which the results would not be impaired by the overwhelming proportion of mRNA reads that is normally seen in RNA-seq (reclassifying lncRNAs downloaded from PLncDB does not prove that NCodR can really add information to an unannotated dataset).

Additional note that shouldn’t affect the editor decision about this paper:

In the Docker image, NCodR could benefit from being parallelized, for time performances, in a future version.

---

## [Reviewer Report]

*Comments to Author*: The reviewer has tried the software on a Physcomitrella RNAseq dataset and has found, rather worryingly, that only a tiny proportion of the sequences therein are classified as mRNA (with the expectation being that an RNAseq dataset for a gene expression paper would contain lots of mRNA). They suggest that the authors try the approach on a similar dataset; and I further suggest that they explain this apparently odd result in an example in the paper.

Perhaps the training on large, diverse ncRNA sets biases the approach to classify even mRNAs as different types of ncRNA? If so, this would challenge the application that the reviewer suggests -- sorting RNAseq data into different categories. If the use of this tool is more specialist -- ie applicable only to sets that are known to be ncRNA of different types -- this should be clarified in the ms.

Following their recommendation, I suggest that this is a potential major revision -- though if there is a simple fix this could easily be minor.

Some smaller comments from me:

Fig 2E -- a comparison between the authors' classifier and others -- has been included. Why are nRC and ncRDeep so bad? Their TPR is barely over 50% and correspondingly their FNR, FDR approach 50%. How does this performance compare with that reported in their original publications? If there is a discrepancy, why?

Please give the definitions of the various ncRNA types in the introduction paragraph where their abbreviations come up.

---

## [Reviewer Report]

August 22nd, 2022

To,

Dr. Olivier Hamant

Editor-in-Chief,

Quantitative Plant Biology

Dear Prof. Hamant,

Thank you very much for your email dated June 24, 2022 attaching the referees’ comments on our manuscript entitled “NCodR: A multi-class SVM classification to distinguish between non-coding RNAs in Viridiplantae” for revision. We thank the associate editor and the reviewers for appreciating our work and for raising careful concerns towards the betterment of the study. I believe that incorporating these suggestions definitely improves the quality of the manuscript and hope Quantitative Plant Biology will accept it. Following, we dissected the referee’s comments and the relevant modifications/additions are coluored in red in the manuscript.

With regards,

Ranjit P. Bahadur,

Professor, Computational Structural Biology Lab

Department of Biotechnology

Indian Institute of Technology Kharagpur

Kharagpur-721302, India

---

## [Reviewer Report]

*Comments to Author*: Thanks for addressing the previous round of comments from me and the reviewer. To my eyes the manuscript now positions itself appropriately as an nc-focussed classification tool with an explanation that it is not designed for higher-level RNA classification. The odd performance of some other approaches is also addressed (perhaps the authors would like to contact the authors of those other approaches to point out those problems for plant RNAs?). I am happy to recommend acceptance of the manuscript in this form.